# Fourth-order kinematic analysis: Advanced decomposition methods for particle motion in modified orthogonal frame

**Fatimah Alghamdi[1]⊕\*, Ayman Elsharkawy[2]⊕\***

**1** Department of Mathematics and Statistics, College of Science, University of Jeddah, Jeddah, 21589, Saudi Arabia, **2** Department of Mathematics, Faculty of Science, Tanta University, Tanta, Egypt

⊕ These authors contributed equally to this work.
\* fmalghamdi@uj.edu.sa (FA); ayman_ramadan@science.tanta.edu.eg (AE)

## Abstract

The analysis of higher-order derivatives of position, particularly the snap (fourth derivative), provides critical insights into complex motion dynamics in fields such as robotics, aerospace, and biomechanics. This research investigates novel decomposition techniques for the fourth-order derivative of position (snap vector) of a particle traveling along spatial curves within three-dimensional Euclidean geometry. We establish comprehensive analytical expressions that decompose the snap vector into components corresponding to modified orthonormal basis vectors. Furthermore, we present an innovative alternative formulation that partitions the snap vector into tangential and radial components within the osculating and rectifying planes. The developed theoretical framework is validated through computational examples, illustrating the practical implementation and effectiveness of the proposed decomposition methodologies.

## 1 Introduction

Understanding complex physical phenomena requires interdisciplinary approaches that bridge theoretical mathematics and applied physics. The investigation of particle motion, particularly in three-dimensional space, represents a fundamental challenge that has driven advances in both differential geometry and classical mechanics. The intricate relationship between mathematical frameworks and physical reality becomes particularly evident when examining higher-order derivatives of position vectors, where traditional analytical methods often prove insufficient for comprehensive understanding.

Classical mechanics, as a cornerstone of physics, examines the motion of objects under various force conditions and their environmental interactions. This field traditionally divides into kinematics, which describes motion patterns, and dynamics, which investigates the underlying causes of motion. Within this framework, the study of higher-order derivatives beyond acceleration has gained increasing importance

**Data availability statement:** All relevant data are within the manuscript.

**Funding:** This work was supported and funded by the Deputyship for Research & Innovation, Ministry of Education in Saudi Arabia, for funding this research work through the project number MoE-IF-UJ-R2-22-04103029-1.

**Competing interests:** The authors declare that they have no known financial conflicts of interest or personal relationships that could have influenced the work presented in this paper.

in modern applications, particularly in understanding complex motion patterns that occur in advanced engineering systems and natural phenomena.

The mathematical foundation for analyzing particle trajectories relies heavily on differential geometry, which employs sophisticated tools from calculus, linear algebra, and multilinear algebra to examine geometric problems. The synergy between differential geometry and particle kinematics becomes particularly pronounced when investigating the geometric properties of trajectories in three-dimensional Euclidean space, where each moving particle with constant mass follows a path characterized by its associated moving frame.

Higher-order derivatives of motion, specifically jerk and snap, represent critical parameters that are often overlooked in traditional mechanical analysis. These quantities, representing the third and fourth derivatives of position with respect to time, play crucial roles in various practical applications. The significance of these parameters becomes evident in situations involving rapid transitions, vibration analysis, and precision control systems where smooth motion profiles are essential.

The mathematical understanding of these higher-order derivatives reveals important insights into motion characteristics. When velocity changes occur gradually from zero, the presence of non-zero jerk becomes apparent. Similarly, when jerk itself changes gradually from zero, the fourth derivative (snap) becomes significant. This pattern continues through higher-order derivatives, each providing additional information about the motion's complexity and smoothness.

Applications of jerk and snap analysis span numerous fields, including physics, engineering, and manufacturing. In vibration analysis, these parameters help identify multi-resonant modes and transition behaviors [1–3]. Civil engineering applications include railway and roadway transition design, where smooth acceleration profiles are crucial for passenger comfort and structural integrity. Industrial applications encompass oscillator design, precision manufacturing, and advanced motion control systems.

The historical development of acceleration decomposition methods provides important context for current research. Siacci [4] pioneered the decomposition of acceleration vectors into tangential and normal components, establishing fundamental principles that continue to influence modern analysis. While Siacci's theorem represented a significant breakthrough, subsequent researchers identified areas for improvement in both formulation and proof methodology.

Whittaker [5] provided enhanced geometrical proofs of Siacci's theorem in 1944, followed by contributions from Grossman [6]. These developments established more rigorous mathematical foundations for vector decomposition techniques. Further extensions of Siacci's theorem were explored by Casey [7], Küçükarslan et al. [8], Özen et al. [9], and Özen [10], who investigated applications to various curve and surface geometries.

Research into jerk vector decomposition has been advanced by several scholars, including Résal [11], Özen et al. [12], Güner [13], and Tosun and Hızarcıoglu [14], who developed resolution methods for different frame types including Frenet, modified, Bishop, and Darboux frames in three-dimensional Euclidean space. Recent

contributions by Hamouda et al. [15] extended these investigations to include both jerk and snap vector resolutions for quasi curves in $\mathbb{E}^3$. Additional relevant studies can be found in [16–19].

Moving frames represent essential tools in classical differential geometry, providing the capability to traverse space curves while revealing characteristic properties such as curvature and torsion. The Frenet frame consists of three orthonormal vectors: the tangent vector (indicating instantaneous direction of motion), the normal vector (pointing toward the center of curvature), and the binormal vector (orthogonal to both). However, the Frenet frame suffers from a critical limitation: it becomes undefined when curves exhibit vanishing curvature, even temporarily. This limitation necessitates alternative approaches that maintain mathematical validity across all curve configurations.

The modified orthogonal frame, introduced by Sasai [20], addresses the limitations of the Frenet frame by providing an alternative orthogonal framework with corresponding formulas analogous to the Frenet equations. This approach offers enhanced mathematical stability and broader applicability for curve analysis. Comprehensive discussions of modified frame applications can be found in [21–23].

This investigation focuses on developing comprehensive snap vector resolution methods for point particles moving along modified curves in three-dimensional Euclidean space. This paper contributes to the presentation and further development of analytical techniques and alternative decomposition approaches in higher-order kinematic analysis. The following sections present: fundamental concepts of modified frames and their relationship to Frenet frames (Sect 2), derivation of snap vector resolutions using modified frames and alternative tangential-radial decompositions (Sect 3), illustrative examples demonstrating practical applications (Sect 4), and comprehensive conclusions summarizing our contributions (Sect 5).

## 2 Preliminaries

This section builds upon the methodology and analytical framework developed by Hamouda et al. [15] for higher-order kinematic analysis. While following similar decomposition approaches, our work extends their methodology by employing modified orthonormal frames instead of quasi-frames, resulting in distinct parameter designations and formulation differences. The key advantages of our approach include enhanced mathematical stability when curvature vanishes and alternative geometric interpretations through tangential-radial decompositions.

This section establishes the fundamental mathematical framework and notation necessary for our subsequent analysis. We review key concepts from differential geometry, including curve parameterization, Frenet frames, and modified orthonormal frames, which form the foundation for our decomposition methods.

Assume that $\mathbb{E}^3$ supplied with the Cartesian metric

$$g = dx_1^2 + dx_2^2 + dx_3^2,$$

where $(x_1, x_2, x_3)$ is a coordinate system of $\mathbb{E}^3$. For any two vectors $P = (p_1, p_2, p_3)$ and $Q = (q_1, q_2, q_3)$ in $\mathbb{E}^3$, the inner product and vector cross product are defined as:

$$\langle P, Q \rangle = p_1 q_1 + p_2 q_2 + p_3 q_3,$$

and

$$P \times Q = \begin{vmatrix} e_1 & e_2 & e_3 \\ p_1 & p_2 & p_3 \\ q_1 & q_2 & q_3 \end{vmatrix} = (p_2 q_3 - p_3 q_2, p_3 q_1 - p_1 q_3, p_1 q_2 - p_2 q_1),$$

respectively [24]. Also, *Cartesian norm* defined by $\|Q\| = \sqrt{\langle Q, Q \rangle}$.

- **Regular curve:** A differentiable curve $\gamma(u)$ in $\mathbb{E}^3$ with $\gamma'(u) \neq 0$ for all $u$.
- **Arc-length parameterized curve:** A regular curve with $\|\gamma'(u)\| = 1$ for all $u$ [24].

Assume that $\gamma(u)$ is a unit speed curve in $\mathbb{E}^3$ with $\kappa \neq 0$, and that

$$
\begin{aligned}
\mathbf{T}(u): &\quad \text{unit tangent vector;} \\
\mathbf{N}(u): &\quad \text{Frenet-normal vector;} \\
\mathbf{B}(u): &\quad \text{Frenet-binormal vector.}
\end{aligned}
$$

The *Frenet orthonormal frame* along $\gamma(u)$ is defined as:

$$
\begin{aligned}
\mathbf{T}(u) &= \gamma'(u); \\
\mathbf{N}(u) &= \mathbf{T}'(u) / \|\mathbf{T}'(u)\|; \\
\mathbf{B}(u) &= \mathbf{T}(u) \times \mathbf{N}(u).
\end{aligned}
\tag{2.1}
$$

Then, the *Frenet equations* for $\gamma(u)$ are

$$
\begin{bmatrix} \mathbf{T}'(u) \\ \mathbf{N}'(u) \\ \mathbf{B}'(u) \end{bmatrix} = \begin{bmatrix} 0 & \kappa & 0 \\ -\kappa & 0 & \tau \\ 0 & -\tau & 0 \end{bmatrix} \begin{bmatrix} \mathbf{T}(u) \\ \mathbf{N}(u) \\ \mathbf{B}(u) \end{bmatrix},
\tag{2.2}
$$

where $\kappa(u) = \|\mathbf{T}'(u)\|$ and $\tau(u) = -\langle \mathbf{B}'(u), \mathbf{N}(u) \rangle$ are Frenet-curvatures functions for $\gamma(u)$.

The *modified orthogonal frame* along $\gamma(u)$ is defined as:

$$
\begin{aligned}
\mathbf{T}(u) &= \gamma'(u); \\
\mathbf{N_m}(u) &= \mathbf{T}'(u) \ \text{(modified-normal vector);} \\
\mathbf{B_m}(u) &= \mathbf{T}(u) \times \mathbf{N_m}(u) \ \text{(modified-binormal vectors).}
\end{aligned}
\tag{2.3}
$$

For the relationship between the modified frame and Frenet frame, the *relation matrix* along $\gamma(u)$ is given by:

$$
\begin{bmatrix} \mathbf{T}(u) \\ \mathbf{N_m}(u) \\ \mathbf{B_m}(u) \end{bmatrix} = \begin{bmatrix} 1 & 0 & 0 \\ 0 & \kappa & 0 \\ 0 & 0 & \kappa \end{bmatrix} \begin{bmatrix} \mathbf{T}(u) \\ \mathbf{N}(u) \\ \mathbf{B}(u) \end{bmatrix},
\tag{2.4}
$$

and so

$$
\begin{bmatrix} \mathbf{T}(u) \\ \mathbf{N}(u) \\ \mathbf{B}(u) \end{bmatrix} = \begin{bmatrix} 1 & 0 & 0 \\ 0 & 1/\kappa & 0 \\ 0 & 0 & 1/\kappa \end{bmatrix} \begin{bmatrix} \mathbf{T}(u) \\ \mathbf{N_m}(u) \\ \mathbf{B_m}(u) \end{bmatrix},
\tag{2.5}
$$

From (2.2)–(2.5), the *modified equations* for $\gamma(u)$ are

$$
\begin{bmatrix} \mathbf{T}'(u) \\ \mathbf{N_m'}(u) \\ \mathbf{B_m'}(u) \end{bmatrix} = \begin{bmatrix} 0 & 1 & 0 \\ -\kappa^2 & \kappa'/\kappa & \tau \\ 0 & -\tau & \kappa'/\kappa \end{bmatrix} \begin{bmatrix} \mathbf{T}(u) \\ \mathbf{N_m}(u) \\ \mathbf{B_m}(u) \end{bmatrix}.
\tag{2.6}
$$

The modified orthogonal frame offers a significant advantage over the Frenet frame in handling curves with vanishing curvature. While the Frenet frame becomes undefined when $\kappa = 0$ (as $\mathbf{N} = \mathbf{T}'/\|\mathbf{T}'\|$ becomes indeterminate), the modified frame vectors $\mathbf{N_m} = \mathbf{T}'$ and $\mathbf{B_m} = \mathbf{T} \times \mathbf{T}'$ remain mathematically valid for all regular curves regardless of curvature values. This enhanced stability makes the modified frame particularly valuable for analyzing curves with inflection points or segments of zero curvature. It is important to note that Eqs (2.4), (2.5), and (2.6) are derived under the assumption $\kappa \neq 0$ to maintain the relationship between modified and Frenet frames. However, the modified frame itself remains well-defined and can be used for kinematic analysis even when $\kappa = 0$, with appropriate adjustments to the derivative formulas.

**Definition 2.1.** *The Snap (jounce) vector is the fourth derivative of the position vector relative to time.*

## 2.1 Notation

During the main results, we need the following notations:

$$
\begin{aligned}
\mathcal{P} &: \quad \text{A point particle} \\
\mathcal{O} &: \quad \text{The fixed origin} \\
\pi &: \quad \text{The osculating plane} \\
\pi^* &: \quad \text{The rectifying plane} \\
H &: \quad \text{The foot of perpendicular line from } \mathcal{O} \text{ to } \pi \\
G &: \quad \text{The foot of perpendicular line from } \mathcal{O} \text{ to } \pi^* \\
H\mathcal{P} &: \quad \text{The radial direction in the plane } \pi \\
G\mathcal{P} &: \quad \text{The radial direction in the plane } \pi^* \\
\mathbf{e_r} &: \quad \text{The unit vector in direction } H\mathcal{P} \\
\mathbf{e_{r^*}} &: \quad \text{The unit vector in direction } G\mathcal{P} \\
\mathcal{H}^{\mathcal{O}} &: \quad \text{The vector of angular momentum of } \mathcal{P} \text{ around } \mathcal{O}.
\end{aligned}
$$

## 3 Main results

Building upon the decomposition techniques introduced by Hamouda et al. [15], this section presents our principal theoretical contributions with modified orthonormal frames. The key differences from prior work include: the use of modified frames rather than quasi-frames or Frenet frames, distinct parameter definitions leading to variations in the resulting coefficients, and an alternative tangential-radial decomposition approach that provides new geometric insights into snap vector components.

In this section, we present our principal theoretical contributions: two novel decomposition theorems for the snap vector. The first theorem provides a decomposition using the modified orthonormal frame, while the second theorem offers an alternative formulation using tangential and radial components. We also include a corollary addressing the special case of planar motion.

**Theorem 3.1.** *Let $\gamma(u)$ be a unit speed modified curve in $\mathbb{E}^3$, $\mathcal{P}$ with mass $m$ (a constant) moves along $\gamma(u)$ and the arc-length $u$ of $\gamma$ synchronizes with $t$. Then, at time $t$, the snap vector of $\mathcal{P}$ is*

$$\mathfrak{U} = \mathcal{C}_{\mathbf{T}}\mathbf{T} + \mathcal{C}_{\mathbf{N_m}}\mathbf{N_m} + \mathcal{C}_{\mathbf{B_m}}\mathbf{B_m}, \tag{3.1}$$

*where*

$$\mathcal{C}_{\mathbf{T}} \quad : \quad = \left(\frac{\mathrm{d}^4 u}{\mathrm{d}t^4}\right) - 6\kappa^2 \left(\frac{\mathrm{d}u}{\mathrm{d}t}\right)^2 \left(\frac{\mathrm{d}^2 u}{\mathrm{d}t^2}\right) - 3\kappa \left(\frac{\mathrm{d}\kappa}{\mathrm{d}u}\right) \left(\frac{\mathrm{d}u}{\mathrm{d}t}\right)^4,$$

$$\mathcal{C}_{\mathbf{N_m}} \quad : \quad = 4 \left(\frac{\mathrm{d}u}{\mathrm{d}t}\right) \left(\frac{\mathrm{d}^3 u}{\mathrm{d}t^3}\right) + 3 \left(\frac{\mathrm{d}^2 u}{\mathrm{d}t^2}\right)^2 + \frac{6}{\kappa} \left(\frac{\mathrm{d}\kappa}{\mathrm{d}u}\right) \left(\frac{\mathrm{d}u}{\mathrm{d}t}\right)^2 \left(\frac{\mathrm{d}^2 u}{\mathrm{d}t^2}\right)$$

$$+ \frac{1}{\kappa} \left(\frac{\mathrm{d}^2 \kappa}{\mathrm{d}u^2}\right) \left(\frac{\mathrm{d}u}{\mathrm{d}t}\right)^4 - \kappa^2 \left(\frac{\mathrm{d}u}{\mathrm{d}t}\right)^4 - \tau^2 \left(\frac{\mathrm{d}u}{\mathrm{d}t}\right)^4,$$

$$\mathcal{C}_{\mathbf{B_m}} \quad : \quad = 6\tau \left(\frac{\mathrm{d}u}{\mathrm{d}t}\right)^2 \left(\frac{\mathrm{d}^2 u}{\mathrm{d}t^2}\right) + \frac{2\tau}{\kappa} \left(\frac{\mathrm{d}\kappa}{\mathrm{d}u}\right) \left(\frac{\mathrm{d}u}{\mathrm{d}t}\right)^4 + \left(\frac{\mathrm{d}\tau}{\mathrm{d}u}\right) \left(\frac{\mathrm{d}u}{\mathrm{d}t}\right)^4.$$

*Proof*: Assume that $\mathcal{P}$ moves along $\gamma(u)$ (a unit speed modified curve) in $\mathbb{E}^3$. Then, according to the modified frame, $\mathcal{P}$ has a position vector with respect to $\mathcal{O}$ in $\mathbb{E}^3$, let it be $\mathcal{X}$ at time $t$. Since the arc-length $u$ of $\gamma$ synchronize with time $t$, $\mathbf{T}(u)$ is given by

$$\mathbf{T}(u) = \frac{\mathrm{d}\mathcal{X}}{\mathrm{d}u}. \tag{3.2}$$

From (2.6) and (3.2), according to the modified frame, the vectors $\mathcal{V}$, $\mathcal{A}$ and $\mathcal{J}$ are given by

$$\mathcal{V} \quad = \quad \frac{\mathrm{d}\mathcal{X}}{\mathrm{d}t} = \left(\frac{\mathrm{d}u}{\mathrm{d}t}\right) \mathbf{T}, \tag{3.3}$$

$$\mathcal{A} \quad = \quad \left(\frac{\mathrm{d}^2 u}{\mathrm{d}t^2}\right) \mathbf{T} + \left(\frac{\mathrm{d}u}{\mathrm{d}t}\right)^2 \mathbf{N_m},$$

and

$$\mathcal{J} = \left[ \left(\frac{\mathrm{d}^3 u}{\mathrm{d}t^3}\right) - \kappa^2 \left(\frac{\mathrm{d}u}{\mathrm{d}t}\right)^3 \right] \mathbf{T} + \left[ 3 \left(\frac{\mathrm{d}u}{\mathrm{d}t}\right) \left(\frac{\mathrm{d}^2 u}{\mathrm{d}t^2}\right) + \frac{1}{\kappa} \left(\frac{\mathrm{d}\kappa}{\mathrm{d}u}\right) \left(\frac{\mathrm{d}u}{\mathrm{d}t}\right)^3 \right] \mathbf{N_m}$$

$$+ \left[ \tau \left(\frac{\mathrm{d}u}{\mathrm{d}t}\right)^3 \right] \mathbf{B_m},$$

respectively. Hence, the snap vector of $\mathcal{P}$ according to the modified frame is expressed as in (3.1). The proof is complete. □

**Theorem 3.2.** *Let $\gamma(u)$ be a unit speed modified curve in $\mathbb{E}^3$, and $\mathcal{P}$ with mass $m$ (a constant) moves along $\gamma(u)$. Then, the vector of angular momentum of $\mathcal{P}$ around $\mathcal{O}$ is*

$$\mathcal{H}^{\mathcal{O}} = m\mu \left(\frac{\mathrm{d}u}{\mathrm{d}t}\right) \mathbf{N_m} + m\nu \left(\frac{\mathrm{d}u}{\mathrm{d}t}\right) \mathbf{B_m}.$$

*Further, the snap vector of $\mathcal{P}$ is*

$$\mathfrak{U} = \mathcal{F}_{\mathbf{T}} \mathbf{T} + \mathcal{F}_{\mathbf{r}} \mathbf{e_r} + \mathcal{F}_{\mathbf{r}^*} \mathbf{e}, \tag{3.4}$$

*where*

$$\mathcal{F}_{\mathbf{T}} = \left(\frac{\mathrm{d}^4 u}{\mathrm{d}t^4}\right) - 6\kappa^2 \left(\frac{\mathrm{d}u}{\mathrm{d}t}\right)^2 \left(\frac{\mathrm{d}^2 u}{\mathrm{d}t^2}\right) - 3\kappa \left(\frac{\mathrm{d}\kappa}{\mathrm{d}u}\right) \left(\frac{\mathrm{d}u}{\mathrm{d}t}\right)^4 + \frac{4\lambda}{\nu} \left(\frac{\mathrm{d}u}{\mathrm{d}t}\right) \left(\frac{\mathrm{d}^3 u}{\mathrm{d}t^3}\right)$$
$$+ \frac{3\lambda}{\nu} \left(\frac{\mathrm{d}^2 u}{\mathrm{d}t^2}\right)^2 + \frac{6\lambda}{\nu\kappa} \left(\frac{\mathrm{d}\kappa}{\mathrm{d}u}\right) \left(\frac{\mathrm{d}u}{\mathrm{d}t}\right)^2 \left(\frac{\mathrm{d}^2 u}{\mathrm{d}t^2}\right) + \frac{\lambda}{\nu\kappa} \left(\frac{\mathrm{d}^2 \kappa}{\mathrm{d}u^2}\right) \left(\frac{\mathrm{d}u}{\mathrm{d}t}\right)^4$$
$$- \frac{\lambda\kappa^2}{\nu} \left(\frac{\mathrm{d}u}{\mathrm{d}t}\right)^4 - \frac{\lambda\tau^2}{\nu} \left(\frac{\mathrm{d}u}{\mathrm{d}t}\right)^4 - \frac{6\lambda\tau}{\mu} \left(\frac{\mathrm{d}u}{\mathrm{d}t}\right)^2 \left(\frac{\mathrm{d}^2 u}{\mathrm{d}t^2}\right)$$
$$- \frac{2\lambda\tau}{\mu\kappa} \left(\frac{\mathrm{d}\kappa}{\mathrm{d}u}\right) \left(\frac{\mathrm{d}u}{\mathrm{d}t}\right)^4 - \frac{\lambda}{\mu} \left(\frac{\mathrm{d}\tau}{\mathrm{d}u}\right) \left(\frac{\mathrm{d}u}{\mathrm{d}t}\right)^4,$$

$$\mathcal{F}_{\mathbf{r}} = -\frac{4r}{\nu} \left(\frac{\mathrm{d}u}{\mathrm{d}t}\right) \left(\frac{\mathrm{d}^3 u}{\mathrm{d}t^3}\right) - \frac{3r}{\nu} \left(\frac{\mathrm{d}^2 u}{\mathrm{d}t^2}\right)^2 - \frac{6r}{\nu\kappa} \left(\frac{\mathrm{d}\kappa}{\mathrm{d}u}\right) \left(\frac{\mathrm{d}u}{\mathrm{d}t}\right)^2 \left(\frac{\mathrm{d}^2 u}{\mathrm{d}t^2}\right)$$
$$- \frac{r}{\nu\kappa} \left(\frac{\mathrm{d}^2 \kappa}{\mathrm{d}u^2}\right) \left(\frac{\mathrm{d}u}{\mathrm{d}t}\right)^4 + \frac{r\kappa^2}{\nu} \left(\frac{\mathrm{d}u}{\mathrm{d}t}\right)^4 + \frac{r\tau^2}{\nu} \left(\frac{\mathrm{d}u}{\mathrm{d}t}\right)^4,$$

$$\mathcal{F}_{\mathbf{r}^*} = \frac{6\tau r^*}{\mu} \left(\frac{\mathrm{d}u}{\mathrm{d}t}\right)^2 \left(\frac{\mathrm{d}^2 u}{\mathrm{d}t^2}\right) + \frac{2\tau r^*}{\mu\kappa} \left(\frac{\mathrm{d}\kappa}{\mathrm{d}u}\right) \left(\frac{\mathrm{d}u}{\mathrm{d}t}\right)^4 + \frac{r^*}{\mu} \left(\frac{\mathrm{d}\tau}{\mathrm{d}u}\right) \left(\frac{\mathrm{d}u}{\mathrm{d}t}\right)^4.$$

*Proof*: Assume that $\mathcal{P}$ in Fig 1 moves along $\gamma(u)$ (a unit speed modified curve) in $\mathbb{E}^3$. Then, in accordance with the modified frame, we assume $\mathcal{X}$ is the vector of position of $\mathcal{P}$ with respect to $\mathcal{O}$ in $\mathbb{E}^3$. $\mathcal{X}$ is given by

$$\mathcal{X} = \lambda\mathbf{T} - \nu\mathbf{N_m} + \mu\mathbf{B_m}, \tag{3.5}$$

where

$$\lambda = \langle \mathcal{X}, \mathbf{T} \rangle, \ \nu = -\frac{1}{\kappa^2} \langle \mathcal{X}, \mathbf{N_m} \rangle \ \text{and} \ \mu = \frac{1}{\kappa^2} \langle \mathcal{X}, \mathbf{B_m} \rangle. \tag{3.6}$$

We define **r** and **r**$^*$ by

$$\mathbf{r} = \lambda\mathbf{T} - \nu\mathbf{N_m} \ \text{and} \ \mathbf{r}^* = \lambda\mathbf{T} + \mu\mathbf{B_m}. \tag{3.7}$$

If the Cartesian norms of **r** and **r**$^*$ are $r$ and $r^*$, respectively, then

$$r^2 = \langle \mathbf{r}, \mathbf{r} \rangle = \lambda^2 + \kappa^2 \nu^2 \ \text{and} \ (r^*)^2 = \langle \mathbf{r}^*, \mathbf{r}^* \rangle = \lambda^2 + \kappa^2 \mu^2. \tag{3.8}$$

Now, we have

$$\mathcal{H}^{\mathcal{O}} = \mathcal{X} \times m\mathcal{V}.$$

Thus, from (3.3) and (3.5), we get

$$\mathcal{H}^{\mathcal{O}} = m\mu \left(\frac{\mathrm{d}u}{\mathrm{d}t}\right) \mathbf{N_m} + m\nu \left(\frac{\mathrm{d}u}{\mathrm{d}t}\right) \mathbf{B_m}.$$

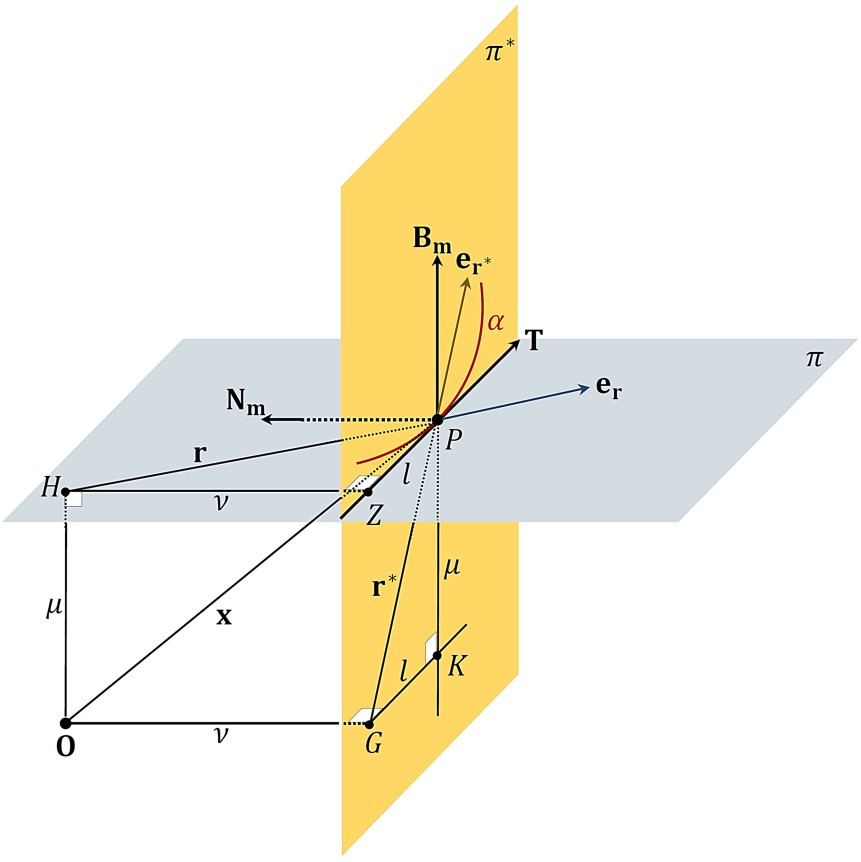

**Fig 1**. **Geometric representation of modified orthonormal frame and decomposition planes.** Motion of particle $\mathcal{P}$ along curve $\gamma$ in Euclidean space $\mathbb{E}^3$, showing the modified orthonormal frame vectors ($\mathbf{T}$, $\mathbf{N_m}$, $\mathbf{B_m}$), osculating plane $\pi$, rectifying plane $\pi^*$, and radial direction vectors ($\mathbf{e_r}$, $\mathbf{e_{r^*}}$) used in snap vector decomposition.

Next, we will write the vectors $\mathbf{N_m}$ and $\mathbf{B_m}$ in terms of $\{\mathbf{r}, \mathbf{T}\}$ and $\{\mathbf{r^*}, \mathbf{T}\}$, respectively, to resolve the snap vector in (3.1) along $\mathbf{T}$, $\mathbf{e_r}$ and $\mathbf{e_{r^*}}$. For this, we use (3.7) in the case where $\nu \neq 0$ and $\mu \neq 0$. We can ensure that $\nu \neq 0$ and $\mu \neq 0$. Thus, from (3.7), we obtain

$$\mathbf{N_m} = \frac{1}{\nu}\left(-\mathbf{r} + \lambda\mathbf{T}\right) \text{ and } \mathbf{B_m} = \frac{1}{\mu}\left(-\lambda\mathbf{T} + \mathbf{r^*}\right). \tag{3.9}$$

Also, from (3.8), we get that $r \neq 0 \neq r^*$. Therefore, $\mathbf{e_r}$ and $\mathbf{e_{r^*}}$ given by

$$\mathbf{e_r} = \frac{1}{r}\mathbf{r} \text{ and } \mathbf{e_{r^*}} = \frac{1}{r^*}\mathbf{r^*}. \tag{3.10}$$

Thus, (3.9) becomes

$$\mathbf{N_m} = \frac{1}{\nu}\left(-r\mathbf{e_r} + \lambda\mathbf{T}\right) \text{ and } \mathbf{B_m} = \frac{1}{\mu}\left(-\lambda\mathbf{T} + r^*\mathbf{e_{r^*}}\right). \tag{3.11}$$

By substituting (3.11) in (3.1), the snap vector of $\mathcal{P}$ is expressed as in (3.4). The proof is complete. □

**Corollary 3.1.** In $\mathbb{E}^3$, Let a unit speed modified curve $\gamma$ traced out by $\mathcal{P}$ is restricted to the plane $\pi$. Assume that the modified-binormal component of $\mathcal{H}^\mathcal{O}$ never vanishes. Hence, the components of $\mathfrak{U}$ become

$$\mathcal{F}_\mathbf{T} = \left(\frac{\mathrm{d}^4 u}{\mathrm{d}t^4}\right) - 6\kappa^2 \left(\frac{\mathrm{d}u}{\mathrm{d}t}\right)^2 \left(\frac{\mathrm{d}^2 u}{\mathrm{d}t^2}\right) - 3\kappa \left(\frac{\mathrm{d}\kappa}{\mathrm{d}u}\right) \left(\frac{\mathrm{d}u}{\mathrm{d}t}\right)^4 + \frac{4\lambda}{\nu} \left(\frac{\mathrm{d}u}{\mathrm{d}t}\right) \left(\frac{\mathrm{d}^3 u}{\mathrm{d}t^3}\right)$$

$$+ \frac{3\lambda}{\nu} \left(\frac{\mathrm{d}^2 u}{\mathrm{d}t^2}\right)^2 + \frac{6\lambda}{\nu\kappa} \left(\frac{\mathrm{d}\kappa}{\mathrm{d}u}\right) \left(\frac{\mathrm{d}u}{\mathrm{d}t}\right)^2 \left(\frac{\mathrm{d}^2 u}{\mathrm{d}t^2}\right) + \frac{\lambda}{\nu\kappa} \left(\frac{\mathrm{d}^2\kappa}{\mathrm{d}u^2}\right) \left(\frac{\mathrm{d}u}{\mathrm{d}t}\right)^4$$

$$- \frac{\lambda\kappa^2}{\nu} \left(\frac{\mathrm{d}u}{\mathrm{d}t}\right)^4 ,$$

$$\mathcal{F}_\mathbf{r} = -\frac{4r}{\nu} \left(\frac{\mathrm{d}u}{\mathrm{d}t}\right) \left(\frac{\mathrm{d}^3 u}{\mathrm{d}t^3}\right) - \frac{3r}{\nu} \left(\frac{\mathrm{d}^2 u}{\mathrm{d}t^2}\right)^2 - \frac{6r}{\nu\kappa} \left(\frac{\mathrm{d}\kappa}{\mathrm{d}u}\right) \left(\frac{\mathrm{d}u}{\mathrm{d}t}\right)^2 \left(\frac{\mathrm{d}^2 u}{\mathrm{d}t^2}\right)$$

$$- \frac{r}{\nu\kappa} \left(\frac{\mathrm{d}^2\kappa}{\mathrm{d}u^2}\right) \left(\frac{\mathrm{d}u}{\mathrm{d}t}\right)^4 + \frac{r\kappa^2}{\nu} \left(\frac{\mathrm{d}u}{\mathrm{d}t}\right)^4 ,$$

$$\mathcal{F}_{\mathbf{r}^*} = 0.$$

$$(3.12)$$

*Proof*: Assume that $\mathcal{P}$ moves along $\gamma(u)$ as shown in Fig 1 that lies in $\pi$ and let $\mathcal{O}$ be a fixed origin in $\mathbb{E}^3$. Then, we have 2-cases. We suppose that $\pi$ does not pass through $\mathcal{O}$. Then, $\mu \neq 0$ and $\nu \neq 0$. In the planar motion, we know that $\tau = 0$. Then, $\mathbf{B_m}$ is constant and vertical to $\pi$. So, $\mu \neq 0$ is a constant, see Fig 2. Consequently, we find from (3.4) that components of are expressed as in (3.12). We suppose that $\pi$ pass through $\mathcal{O}$. Hence, $\mu = 0$. Also, $\nu \neq 0$ and $\tau = 0$. Then,

$$-\frac{\lambda\tau^2}{\nu} \left(\frac{\mathrm{d}u}{\mathrm{d}t}\right)^4$$

and

$$\frac{r\tau^2}{\nu} \left(\frac{\mathrm{d}u}{\mathrm{d}t}\right)^4$$

vanish. While,

$$-\frac{6\lambda\tau}{\mu} \left(\frac{\mathrm{d}u}{\mathrm{d}t}\right)^2 \left(\frac{\mathrm{d}^2 u}{\mathrm{d}t^2}\right), \quad 6\frac{r^*\tau}{\mu} \left(\frac{\mathrm{d}u}{\mathrm{d}t}\right)^2 \left(\frac{\mathrm{d}^2 u}{\mathrm{d}t^2}\right),$$

$$-2\frac{\lambda\tau}{\mu\kappa} \left(\frac{\mathrm{d}\kappa}{\mathrm{d}u}\right) \left(\frac{\mathrm{d}u}{\mathrm{d}t}\right)^4, \quad 2\frac{\tau r^*}{\mu\kappa} \left(\frac{\mathrm{d}\kappa}{\mathrm{d}u}\right) \left(\frac{\mathrm{d}u}{\mathrm{d}t}\right)^4,$$

$$-\frac{\lambda}{\mu} \left(\frac{\mathrm{d}\tau}{\mathrm{d}u}\right) \left(\frac{\mathrm{d}u}{\mathrm{d}t}\right)^4 \text{ and } \frac{r^*}{\mu} \left(\frac{\mathrm{d}\tau}{\mathrm{d}u}\right) \left(\frac{\mathrm{d}u}{\mathrm{d}t}\right)^4$$

have indefiniteness 0/0. So, when $\mu \longrightarrow 0$, we shall discuss this case. Thus, using (3.7), (3.8) and (3.10), we obtain that $r^* \approx \lambda$ and $\mathbf{T} \approx \mathbf{e_{r^*}}$ (see Fig 2). Consequently, the vector

$$\left[-6\frac{\lambda\tau}{\mu} \left(\frac{\mathrm{d}u}{\mathrm{d}t}\right)^2 \left(\frac{\mathrm{d}^2 u}{\mathrm{d}t^2}\right) - 2\frac{\lambda}{\mu}\frac{\tau}{\kappa} \left(\frac{\mathrm{d}\kappa}{\mathrm{d}u}\right) \left(\frac{\mathrm{d}u}{\mathrm{d}t}\right)^4 - \frac{\lambda}{\mu} \left(\frac{\mathrm{d}\tau}{\mathrm{d}u}\right) \left(\frac{\mathrm{d}u}{\mathrm{d}t}\right)^4\right] \mathbf{T}$$

$$+ \left[6\frac{r^*\tau}{\mu} \left(\frac{\mathrm{d}u}{\mathrm{d}t}\right)^2 \left(\frac{\mathrm{d}^2 u}{\mathrm{d}t^2}\right) + 2\frac{r^*}{\mu}\frac{\tau}{\kappa} \left(\frac{\mathrm{d}\kappa}{\mathrm{d}u}\right) \left(\frac{\mathrm{d}u}{\mathrm{d}t}\right)^4 + \frac{r^*}{\mu} \left(\frac{\mathrm{d}\tau}{\mathrm{d}u}\right) \left(\frac{\mathrm{d}u}{\mathrm{d}t}\right)^4\right] \mathbf{e_{r^*}},$$

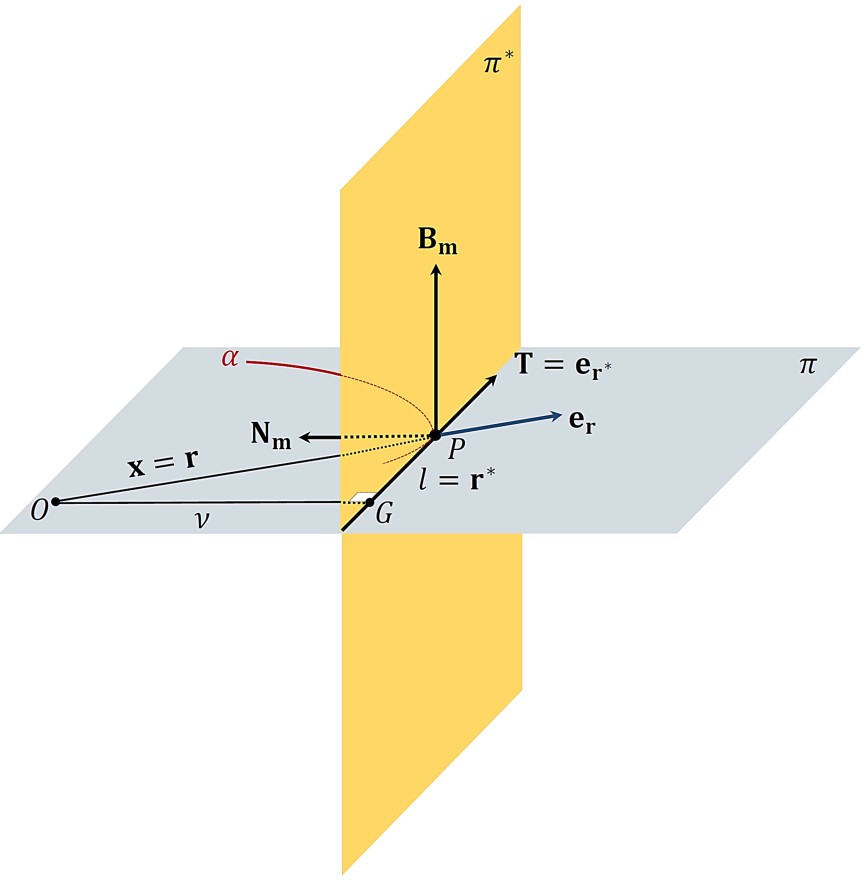

**Fig 2. Planar motion with origin in osculating plane.** The motion of particle $\mathcal{P}$ along curve $\gamma$ when the osculating plane $\pi$ contains the fixed origin $\mathcal{O}$, illustrating the special case analyzed in Corollary 3.1 where $\mu = 0$ and the rectifying plane component vanishes.

coincides with the zero vector at $\mu \longrightarrow 0$. Then, it follows from (3.4) that components are expressed as in (3.12). The proof is complete. $\qquad\square$

## 4 Application

In this section, we provide an illustrative example based on a helical curve, following a similar approach to Guner [13]. While Guner analyzed jerk vector decomposition using Frenet frames, our work extends this example to snap vector decomposition using modified orthonormal frames, demonstrating the application of our theoretical results. We focus on fourth-order rather than third-order derivatives, using the modified frame components instead of Frenet components. Further, an alternative decomposition formulations that provide complementary geometric interpretations are introduced.

**Example 4.1.** Assume that $\mathcal{P}$ is a particle point moving along the helix curve in Fig 3

$$\beta(t) = \left(3\cos\frac{t}{5}, 3\sin\frac{t}{5}, 4\frac{t}{5}\right),$$

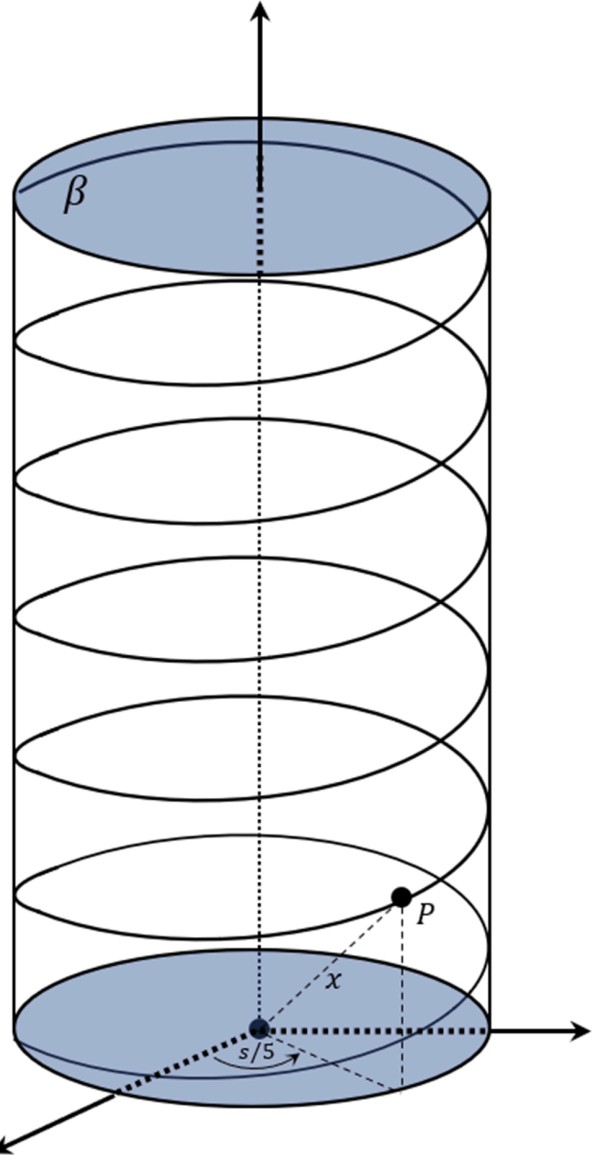

**Fig 3**. **Helical trajectory in three-dimensional space.** The motion of particle $\mathcal{P}$ along the helix curve $\beta(t) = (3\cos\frac{t}{5}, 3\sin\frac{t}{5}, 4\frac{t}{5})$ in $\mathbb{E}^3$, demonstrating the application of snap vector decomposition methods to a curve with constant curvature and torsion.

in Euclidean 3-space $\mathbb{E}^3$. Then, $\mathcal{X}$ of $\mathcal{P}$ at time $t$ is given by

$$\mathcal{X} = \left(3\cos\frac{t}{5}, 3\sin\frac{t}{5}, 4\frac{t}{5}\right). \tag{4.1}$$

By differentiating (4.1) 4-times with respect to time $t$, we get

$$\mathcal{U} = \left(\frac{3}{625}\cos\frac{t}{5}, \frac{3}{625}\sin\frac{t}{5}, 0\right).$$

Since $\beta$ is a unit speed curve, it is clear that

$$\frac{\mathrm{d}u}{\mathrm{d}t} = 1 \text{ and } \frac{\mathrm{d}^2 u}{\mathrm{d}t^2} = \frac{\mathrm{d}^3 u}{\mathrm{d}t^3} = \frac{\mathrm{d}^4 u}{\mathrm{d}t^4} = 0.$$

Thus, we write

$$\beta^*(u) = \left(3\cos\frac{u}{5}, 3\sin\frac{u}{5}, 4\frac{u}{5}\right). \tag{4.2}$$

From (2.3), the modified frame for the helix curve is

$$
\begin{aligned}
\mathbf{T}(u) &= \left(-\frac{3}{5}\sin\frac{u}{5}, \frac{3}{5}\cos\frac{u}{5}, \frac{4}{5}\right), \\
\mathbf{N_m}(u) &= \left(-\frac{3}{25}\cos\frac{u}{5}, -\frac{3}{25}\sin\frac{u}{5}, 0\right), \\
\mathbf{B_m}(u) &= \left(\frac{12}{125}\sin\frac{u}{5}, -\frac{12}{125}\cos\frac{u}{5}, \frac{9}{125}\right).
\end{aligned}
$$

Thus, the Frenet-curvatures are

$$\kappa(u) = \frac{3}{25} \text{ and } \tau(u) = \frac{4}{25}.$$

By means of (3.6), (3.8) and (4.2), we get

$$\lambda = \frac{16}{25}u, \ \nu = 25, \ \mu = \frac{36}{625}u, \ r = \sqrt{\frac{256}{625}u^2 + 9} \text{ and } r^* = 0.64u.$$

By applying Theorem 3.1, we obtain

$$\mathcal{U} = -\frac{1}{25}\mathbf{N_m}.$$

By applying Theorem 3.2, we obtain

$$\mathcal{U} = -\frac{16}{15625}\mathbf{T} + \sqrt{\frac{256}{15625}u^2 + \frac{9}{25}}\mathbf{e_r}.$$

The example demonstrates the practical utility of our theoretical framework in decomposing complex motion characteristics. The helix curve serves as an excellent test case due to its non-zero curvature and torsion, allowing us to validate both decomposition methods. This application highlights how our approach can be implemented computationally and provides insights into motion analysis for engineering applications such as robotic path planning, vehicle dynamics, and animation systems, where higher-order smoothness is crucial.

**Example 4.2.** [25] Consider a curve given by the parametric equation

$$\alpha(s) = \left(\frac{1}{\sqrt{2}}\int_0^s \cos\left(\frac{\pi t^2}{2}\right)dt, \frac{1}{\sqrt{2}}\int_0^s \sin\left(\frac{\pi t^2}{2}\right)dt, \frac{s}{\sqrt{2}}\right), \tag{4.3}$$

which is a *helical curve over a clothoid* (also known as Cornu spiral or Euler spiral) as in Fig 4. This curve has important practical applications in highway and railway route design, roller coaster engineering, and transition curve design, where smooth curvature variation is essential for passenger comfort and safety.

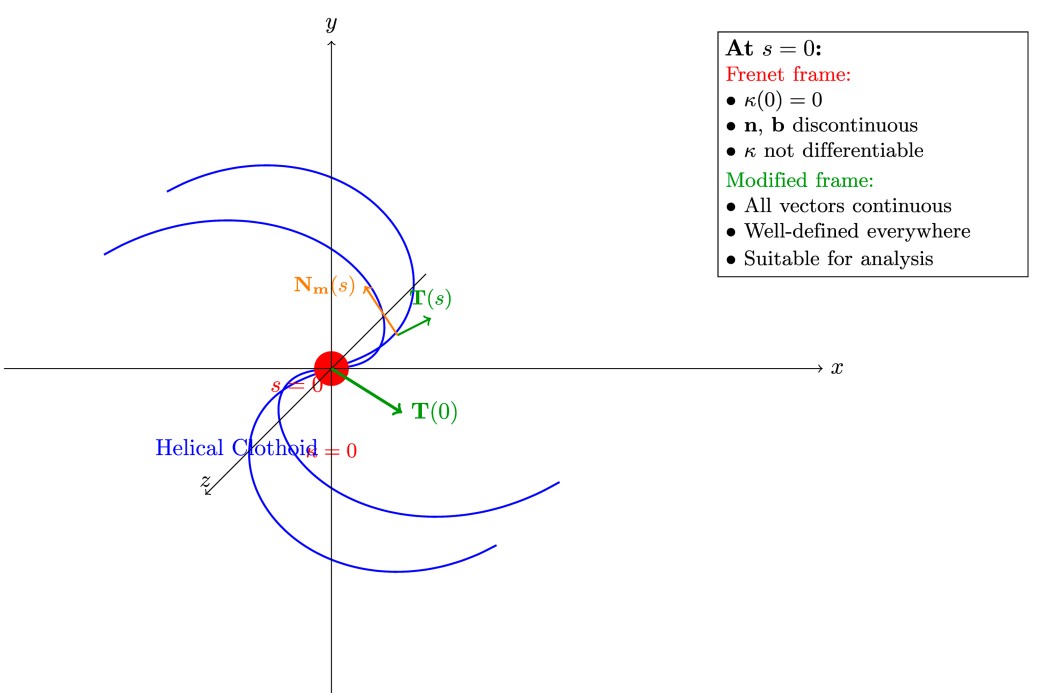

**Fig 4. Helical clothoid curve with vanishing curvature.** Helical curve over clothoid (Euler spiral) in $\mathbb{E}^3$, demonstrating a curve where curvature vanishes at $s = 0$. This example illustrates the critical advantage of the modified orthonormal frame, which remains well-defined at the inflection point where the Frenet frame becomes discontinuous, enabling continuous snap vector analysis throughout the curve.

The components $\int_0^s \cos\left(\frac{\pi t^2}{2}\right) dt$ and $\int_0^s \sin\left(\frac{\pi t^2}{2}\right) dt$ are called *Fresnel integrals*, which appear frequently in diffraction theory and engineering applications.

The Frenet frame of the curve $\alpha$ are:

$$\mathbf{T}(s) = \left(\frac{1}{\sqrt{2}}\cos\left(\frac{\pi s^2}{2}\right), \frac{1}{\sqrt{2}}\sin\left(\frac{\pi s^2}{2}\right), \frac{1}{\sqrt{2}}\right), \tag{4.4}$$

$$\mathbf{N}(s) = \left(-\frac{s}{|s|}\sin\left(\frac{\pi s^2}{2}\right), \frac{s}{|s|}\cos\left(\frac{\pi s^2}{2}\right), 0\right), \tag{4.5}$$

$$\mathbf{B}(s) = \left(-\frac{s}{\sqrt{2}\,|s|}\cos\left(\frac{\pi s^2}{2}\right), -\frac{s}{\sqrt{2}\,|s|}\sin\left(\frac{\pi s^2}{2}\right), \frac{s}{\sqrt{2}\,|s|}\right), \tag{4.6}$$

with curvature

$$\kappa(s) = \frac{\pi\,|s|}{\sqrt{2}}. \tag{4.7}$$

**Critical Problems at $s = 0$:**

At the point $s = 0$, the Frenet frame exhibits severe pathological behavior:

• The curvature $\kappa(0) = 0$ (vanishes).

- The curvature is **not differentiable** at $s = 0$ due to the absolute value $|s|$.
- The principal normal vector $\mathbf{N}(s)$ contains the term $s/|s|$, making it **discontinuous** at $s = 0$.
- Similarly, the binormal vector $\mathbf{B}(s)$ is also **discontinuous** at $s = 0$.

These discontinuities make the Frenet frame completely unsuitable for kinematic analysis at or near $s = 0$, precisely where smooth transition behavior is most critical in engineering applications.

**Modified Orthogonal Frame Analysis:**

In contrast, the modified orthogonal frame of the curve $\alpha$ is given by:

$$\mathbf{T}(s) = \left( \frac{1}{\sqrt{2}} \cos\left( \frac{\pi s^2}{2} \right), \frac{1}{\sqrt{2}} \sin\left( \frac{\pi s^2}{2} \right), \frac{1}{\sqrt{2}} \right), \tag{4.8}$$

$$\mathbf{N_m}(s) = \left( -\frac{\pi s}{\sqrt{2}} \sin\left( \frac{\pi s^2}{2} \right), \frac{\pi s}{\sqrt{2}} \cos\left( \frac{\pi s^2}{2} \right), 0 \right), \tag{4.9}$$

$$\mathbf{B_m}(s) = \left( -\frac{\pi s}{2} \cos\left( \frac{\pi s^2}{2} \right), -\frac{\pi s}{2} \sin\left( \frac{\pi s^2}{2} \right), \frac{\pi s}{2} \right). \tag{4.10}$$

**Crucial Advantages at $s = 0$:**

Evaluating at $s = 0$:

$$\mathbf{T}(0) = \left( \frac{1}{\sqrt{2}}, 0, \frac{1}{\sqrt{2}} \right) \quad \text{(well-defined)} \tag{4.11}$$

$$\mathbf{N_m}(0) = (0, 0, 0) \quad \text{(well-defined as zero vector)} \tag{4.12}$$

$$\mathbf{B_m}(0) = (0, 0, 0) \quad \text{(well-defined as zero vector)} \tag{4.13}$$

Moreover, all three vectors are **continuous** at $s = 0$:

$$\lim_{s \to 0} \mathbf{T}(s) = \mathbf{T}(0), \quad \lim_{s \to 0} \mathbf{N_m}(s) = \mathbf{N_m}(0), \quad \lim_{s \to 0} \mathbf{B_m}(s) = \mathbf{B_m}(0).$$

For a particle $\mathcal{P}$ moving along this helical clothoid, the snap vector can be analyzed using the modified frame throughout the entire curve, including at $s = 0$. Using Theorem 3.1, at any point $s \neq 0$, we can compute:

$$\kappa(s) = \frac{\pi |s|}{\sqrt{2}},$$

$$\frac{d\kappa}{ds} = \frac{\pi s}{\sqrt{2}|s|} = \frac{\pi \operatorname{sgn}(s)}{\sqrt{2}},$$

$$\frac{d^2\kappa}{ds^2} = 0 \quad (s \neq 0),$$

and the snap vector components can be evaluated using Eq (3.1). As $s \to 0$, the continuity of the modified frame ensures that the kinematic analysis remains well-behaved, which is impossible with the Frenet frame approach.

This example justifies our theoretical development and demonstrates why the modified orthogonal frame methodology is essential for higher-order kinematic analysis in realistic engineering scenarios.

## 5 Conclusion

In this work, we have advanced the kinematic analysis of particle motion by deriving novel decomposition methods for the snap vector (fourth derivative of position) using the modified orthogonal frame. Our approach offers several significant advantages over existing methods:

First, the modified orthogonal frame provides mathematical stability that surpasses the traditional Frenet frame, particularly in cases of vanishing curvature where Frenet frames become undefined. This extends the applicability of higher-order kinematic analysis to a broader class of curves and motion patterns.

Second, our dual decomposition approach, both in the modified orthonormal basis and in tangential-radial components, offers complementary perspectives that enhance understanding of complex motion characteristics. The tangential-radial decomposition, in particular, provides intuitive geometric interpretations that can be valuable in practical applications.

Third, the computational example validates our theoretical framework and demonstrates its implementability for real-world problems. The methods developed here show particular promise for applications requiring precise motion control, including robotics, aerospace trajectory planning, and computer animation.

Compared to existing approaches that primarily focus on lower-order derivatives or rely on less stable coordinate systems, our work provides a more comprehensive and robust framework for fourth-order kinematic analysis. Future work may extend these decomposition methods to non-Euclidean spaces, relativistic frameworks, or applications in specific engineering domains where higher-order motion characteristics are critical.

## Author contributions

**Conceptualization:** Ayman Elsharkawy.

**Data curation:** Ayman Elsharkawy.

**Formal analysis:** Fatimah Alghamdi.

**Investigation:** Ayman Elsharkawy.

**Methodology:** Ayman Elsharkawy.

**Project administration:** Fatimah Alghamdi.

**Resources:** Ayman Elsharkawy.

**Software:** Ayman Elsharkawy.

**Supervision:** Fatimah Alghamdi.

**Validation:** Ayman Elsharkawy.

**Visualization:** Ayman Elsharkawy.

**Writing – original draft:** Fatimah Alghamdi, Ayman Elsharkawy.

**Writing – review & editing:** Fatimah Alghamdi, Ayman Elsharkawy.

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
