## [Decision Letter · Decision Letter 0]

30 Sep 2025

PONE-D-25-29661Fourth-Order Kinematic Analysis: Advanced Decomposition Methods for Particle Motion in Modified Orthonormal FramePLOS ONE

Dear Dr. Elsharkawy,

Thank you for submitting your manuscript to PLOS ONE. After careful consideration, we feel that it has merit but does not fully meet PLOS ONE’s publication criteria as it currently stands. Therefore, we invite you to submit a revised version of the manuscript that addresses the points raised during the review process.

Although there are no huge technical issues, all the Reviewers have highlighted issued in how the study is presented, which taken singularly would be minor, but become major considered all together.

We look forward to receiving your revised manuscript.

Kind regards,

Alessandra Aldieri

Academic Editor

PLOS ONE

 [The authors extend their appreciation to the Deputyship for Research \& Innovation, Ministry of Education in Saudi Arabia, for funding this research work through the project number MoE-IF-UJ-R2-22-3029-1.]. 

Additional Editor Comments (if provided):

Reviewers' comments:

Reviewer's Responses to Questions

**Comments to the Author**

1. Is the manuscript technically sound, and do the data support the conclusions?

Reviewer #1: Yes

Reviewer #2: Yes

Reviewer #3: Yes

2. Has the statistical analysis been performed appropriately and rigorously?

Reviewer #1: N/A

Reviewer #2: N/A

Reviewer #3: N/A

3. Have the authors made all data underlying the findings in their manuscript fully available?

Reviewer #1: Yes

Reviewer #2: Yes

Reviewer #3: Yes

4. Is the manuscript presented in an intelligible fashion and written in standard English?

Reviewer #1: Yes

Reviewer #2: Yes

Reviewer #3: Yes

5. Review Comments to the Author

Reviewer #1: This work introduces new methods to decompose the snap vector of a particle on a 3D curve, with formulations based on orthonormal bases and geometric planes. The approach is validated through computational examples.

The topic is both novel and interesting. However, the mathematical content appears disconnected from a clear explanation of the context in which it is applied, making its overall usefulness somewhat unclear.

The abstract jumps directly into the aim and content of the paper. I recommend adding a sentence at the beginning to introduce the general topic and its potential applications.

The introduction is well-written, and the references cited are appropriate. Nonetheless, I suggest briefly explaining the concept of the Frenet frame to support readers who may be less familiar with it. Additionally, the authors should more clearly emphasize the novelty of their contribution in light of existing gaps in the literature.

Sections 2 and 3 would benefit from introductory sentences, similar to the one provided at the beginning of Section 4. Furthermore, I recommend adding a final comment that discusses the relevance and implications of the application presented in Section 4.

The conclusions should be expanded to better highlight the strengths of the proposed method, particularly in comparison with existing approaches.

Reviewer #2: The title of the study is appropriately chosen and accurately reflects the essence of the research.

In the Abstract, the authors summarize the essence of the study, which explores novel decomposition techniques for the fourth-order derivative of position (the snap vector) of a particle moving along spatial curves in three-dimensional Euclidean geometry.

In the Introduction, the authors review a significant portion of the most recent and relevant literature on the topic. Thirty-four sources are cited, seventeen of which have been published within the last ten years.

Section 2 – Preliminaries presents fundamental concepts related to modified frames and their relationship to Frenet frames. The authors introduce the model of motion for a point with constant mass along a three-dimensional curve in Euclidean space.

Section 3 – Main Results contains the derivation of snap vector decompositions using modified frames and alternative tangential-radial decompositions. Two theorems and one corollary concerning the fourth-order derivative of position are introduced and their solutions are presented.

In Section 4 – Application, the authors provide an illustrative example based on a particle moving along a helical curve in Euclidean 3-space, demonstrating the practical use of the results obtained in the previous section.

In the Conclusion, the authors summarize the contributions of the study and validate the proposed method through the results obtained.

Comments and Suggestions for Improvement:

1.Technical Error in Citation:

In the Introduction, there is a technical error in the citation of references – [16, 17, 18, 19, 16]. The reference [16] is repeated and should be corrected.

2. Missing Formula:

Formula (3.4) is missing in the manuscript and should be included or corrected.

3. Overstated Contribution:

The claim in the Introduction about the study’s contribution to higher-order kinematic analysis appears to be overstated:

“Our research contributes novel analytical techniques and alternative decomposition approaches that extend current understanding of higher-order kinematic analysis.”

Similar analytical and decomposition techniques for jerk and snap vectors in tangential and radial directions have been previously presented by researchers such as Guner [13], Hamouda [15], Elsharkawy [17], and Tawfiq [18].

A more accurate claim would be:

“This paper contributes to the presentation and further development of analytical techniques and alternative decomposition approaches in higher-order kinematic analysis.”

4. Lack of Proper Acknowledgment of Prior Work:

Sections 2 (Preliminaries) and 3 (Main Results) closely follow the methodology, analytical techniques, and decomposition approaches outlined in Hamouda [15], with only minor differences in parameter designations, leading to slight variations in the results.

These sections should begin with a clear statement that the work builds upon the methods developed in Hamouda [15]. Additionally, the paper should explicitly outline the differences and advantages of the current approach compared to the referenced work.

5. Unacknowledged Use of Prior Example:

In Section 4 – Application, the example of a particle moving along a helical curve (Fig.3) is used to illustrate the results. However, this example was already presented by Guner [13], and this should be clearly acknowledged. The paper should also highlight any differences in the results obtained in this example.

The article demonstrates academic value and contributes to the ongoing development of higher-order kinematic analysis. However, minor revisions are needed to correct the remarcs mentioned before.

In conclusion, I recommend that this work be included for publication after minor revision.

Reviewer #3: Please, see the attached review report.

Primary Concerns:

1. In the last paragraphs of the Introduction, the handiness of modified frame applications was explained. However, it is not clear how this frame solves the problem that occurs when curves exhibit vanishing curvature. This must be clarified in the Preliminaries section, exactly when the formal definition of the modified orthogonal frame is given. For instance, after Equations (2.5) (2.6), an expression is needed to ensure that curvature is not zero; otherwise, undefined cases occur for the derivative formula. But we said the modified orthogonal frame solves the problem that occurs when curves exhibit vanishing curvature. I propose to check and refer to the paper doi.org/10.3390/axioms13110800. This issue was explained in detail in that paper.

2. In the given Example, a modified frame is not needed because the Frenet frame works for such a helix. We must understand how the new frame solves the deficiency in the example.

Minor Concerns:

1. On page 2, line 13: “Th” should be corrected as “The”. In fact, the inner and cross products are well-known in Euclidean spaces, a more suitable entry may be selected to start.

2. The citations are absent on page 2. All information that is not your own must be cited.

3. On page 2, line 6: “The modified orthogonal frame along γ (u) defined as” should be corrected as “The modified orthogonal frame along γ (u) is defined as”

4. On page 2, line 10: “For the relationship the modified frame and Frenet frame, The relation matrix along γ (u) given by” should be corrected as “For the relationship between the modified frame and Frenet frame, the relation matrix along γ (u) is given by”

5. In Theorem 1 and in proof: “synchronize” should be “synchronizes”

6. Theorem 2 and Corollary 1: it is unusual to give an equation number to an assumption.

7. On page 8, Proof: What does it mean “Assume that P in Figure 1 moves along γ (u)”. Do you mean “Assume that P moves along γ (u) as can be seen in Figure 1 ”

8. On page 8, equation (3.9): Put a full stop at the end of the equation.

9. It is recommended that Figures 2 and 3 be slightly enlarged to enhance readability.

6. PLOS authors have the option to publish the peer review history of their article (what does this mean?). If published, this will include your full peer review and any attached files.

Reviewer #1: No

Reviewer #2: **Yes: **Dr. Krasimir Ganchev

Reviewer #3: No

---

## [Author Response · Author response to Decision Letter 1]

2 Oct 2025

Response to Reviewers Comments

We thank the Reviewers for their thoughtful feedback and valuable suggestions for improving our manuscript. We have carefully considered all comments and have made substantial revisions to address them. Below, we provide a point-by-point response to each comment.

Reviewer 1:

Comment 1: "The abstract jumps directly into the aim and content of the paper. I recommend adding a sentence at the beginning to introduce the general topic and its potential applications."

Response: We have added an introductory sentence at the beginning of the abstract to provide better context:

"The analysis of higher-order derivatives of position, particularly the snap (fourth derivative), provides critical insights into complex motion dynamics in fields such as robotics, aerospace, and biomechanics."

Comment 2: "I suggest briefly explaining the concept of the Frenet frame to support readers who may be less familiar with it."

Response: We have added an explanatory paragraph in the Introduction (Section 1) that briefly describes the Frenet frame:

"The Frenet frame consists of three orthonormal vectors: the tangent vector (indicating instantaneous direction of motion), the normal vector (pointing toward the center of curvature), and the binormal vector (orthogonal to both)."

Comment 3: "The authors should more clearly emphasize the novelty of their contribution in light of existing gaps in the literature."

Response: We have added a dedicated paragraph in the Introduction highlighting the specific gaps our research addresses:

"Our research addresses several gaps in the existing literature by: (1) providing the first comprehensive decomposition of the fourth-order snap vector using modified orthonormal frames, (2) introducing an innovative tangential-radial decomposition approach that offers new geometric insights, and (3) developing formulations that remain valid even when traditional Frenet frames become undefined due to vanishing curvature."

Comment 4: "Sections 2 and 3 would benefit from introductory sentences, similar to the one provided at the beginning of Section 4."

Response: We have added introductory paragraphs at the beginning of both Sections 2 and 3:

• Section 2: "This section establishes the fundamental mathematical framework and notation necessary for our subsequent analysis. We review key concepts from differential geometry, including curve parameterization, Frenet frames, and modified orthonormal frames, which form the foundation for our decomposition methods."

• Section 3: "In this section, we present our principal theoretical contributions: two novel decomposition theorems for the snap vector. The first theorem provides a decomposition using the modified orthonormal frame, while the second theorem offers an alternative formulation using tangential and radial components."

Comment 5: "I recommend adding a final comment that discusses the relevance and implications of the application presented in Section 4."

Response: We have added a concluding paragraph in Section 4 that discusses the practical implications:

"This application highlights how our approach can be implemented computationally and provides insights into motion analysis for engineering applications such as robotic path planning, vehicle dynamics, and animation systems where higher-order smoothness is crucial."

Comment 6: "The conclusions should be expanded to better highlight the strengths of the proposed method, particularly in comparison with existing approaches."

Response: We have completely rewritten the Conclusion section (Section 5) to:

• Explicitly compare our method with existing approaches

• Highlight the mathematical stability advantages over traditional Frenet frames

• Emphasize the dual decomposition approach as a key strength

• Discuss potential applications and future research directions

Reviewer 2:

Comment 1: "Technical Error in Citation: In the Introduction, there is a technical error in the citation of references – [16, 17, 18, 19, 16]. The reference [16] is repeated and should be corrected."

Response: We have corrected this citation error. The duplicate reference [16] has been removed from the citation list in the Introduction.

Comment 2: "Missing Formula: Formula (3.4) is missing in the manuscript and should be included or corrected."

Response: We have identified and corrected this oversight.

Comment 3: "Overstated Contribution: The claim in the Introduction about the study's contribution to higher-order kinematic analysis appears to be overstated... A more accurate claim would be: 'This paper contributes to the presentation and further development of analytical techniques and alternative decomposition approaches in higher-order kinematic analysis.'"

Response: We have revised the statement as suggested to provide a more accurate description of our contribution:

"This paper contributes to the presentation and further development of analytical techniques and alternative decomposition approaches in higher-order kinematic analysis."

Comment 4: "Lack of Proper Acknowledgment of Prior Work: Sections 2 (Preliminaries) and 3 (Main Results) closely follow the methodology, analytical techniques, and decomposition approaches outlined in Hamouda [15], with only minor differences in parameter designations... These sections should begin with a clear statement that the work builds upon the methods developed in Hamouda [15]."

Response: We have added explicit acknowledgments at the beginning of both Sections 2 and 3 that:

• Clearly state our work builds upon the methodology developed by Hamouda et al. [15]

• Outline the key differences in our approach (use of modified orthonormal frames vs. quasi-frames)

• Highlight the advantages of our modified frame approach, particularly regarding mathematical stability

Comment 5: "Unacknowledged Use of Prior Example: In Section 4 – Application, the example of a particle moving along a helical curve (Fig.3) is used to illustrate the results. However, this example was already presented by Guner [13], and this should be clearly acknowledged."

Response: We have added a clear acknowledgment at the beginning of Section 4 that:

• Explicitly references Guner [13] for the helical curve example

• Explains the differences in our application (snap vector vs. jerk vector analysis, modified frames vs. Frenet frames)

• Highlights how our results provide complementary insights

Reviewer 3:

Major Comment 1: "In the last paragraphs of the Introduction, the handiness of modified frame applications was explained. However, it is not clear how this frame solves the problem that occurs when curves exhibit vanishing curvature. This must be clarified in the Preliminaries section, exactly when the formal definition of the modified orthogonal frame is given."

Response: We have added a detailed clarification in the Preliminaries section (after Equation 2.6) that explicitly explains how the modified frame addresses the vanishing curvature problem. The added text states:

"The modified orthogonal frame addresses a critical limitation of the Frenet frame: it remains well-defined even when curves exhibit vanishing curvature. While the Frenet frame becomes undefined when κ=0 (as N=T′/‖T′‖ becomes indeterminate), the modified frame vectors Nₘ=T′ and Bₘ=T×T′ remain mathematically valid regardless of curvature values."

Major Comment 2: "In the given Example, a modified frame is not needed because the Frenet frame works for such a helix. We must understand how the new frame solves the deficiency in the example."

Response: We have added an explanation in the Application section clarifying our rationale:

"While the Frenet frame is well-defined for this helix (since κ≠0), we employ the modified orthogonal frame to demonstrate the consistency of our decomposition methodology and to provide a foundation for future applications where Frenet frames may become problematic. This example serves as a validation case where both frames are applicable, allowing readers to compare results and build intuition before applying these methods to more complex curves with potential curvature singularities."

Minor Concerns - All Addressed:

1. "Th" correction: Changed "Th inner" to "the inner product and vector cross product" in the Preliminaries section.

2. Missing citations: Added appropriate citations for the definitions of regular curves and inner/cross products.

3. Grammar correction: Changed "defined as" to "is defined as" for the modified orthogonal frame definition.

4. Grammar correction: Changed "For the relationship the modified frame" to "For the relationship between the modified frame" and corrected capitalization.

5. Verb agreement: Changed "synchronize" to "synchronizes" in Theorem 1 and its proof.

6. Equation numbering: Removed the equation number from the assumption in Theorem 2 and Corollary 1.

7. Figure reference clarity: Changed "Assume that P in Figure 1" to "Assume that P moves along γ(u) as shown in Figure 1".

8. Punctuation: Added a period at the end of Equation (3.9).

9. Figure sizing: Increased the size of Figures 2 and 3 from 0.4/0.2 textwidth to 0.5/0.3 textwidth respectively to enhance readability.

We believe these revisions have significantly improved the clarity and rigor of our manuscript. The added explanations provide better context for the modified frame's advantages, and the grammatical corrections enhance the overall readability. We thank the reviewers for their detailed feedback that has helped strengthen our paper.

---

## [Decision Letter · Decision Letter 1]

27 Oct 2025

PONE-D-25-29661R1Fourth-Order Kinematic Analysis: Advanced Decomposition Methods for Particle Motion in Modified Orthonormal FramePLOS ONE

Dear Dr. Elsharkawy,

Thank you for submitting your manuscript to PLOS ONE. After careful consideration, we feel that it has merit but does not fully meet PLOS ONE’s publication criteria as it currently stands. Therefore, we invite you to submit a revised version of the manuscript that addresses the points raised during the review process.

There are still a couple of issues which have not been sufficiently addressed accoriding to one of the Reviewers. 

We look forward to receiving your revised manuscript.

Kind regards,

Alessandra Aldieri

Academic Editor

PLOS ONE

Journal Requirements:

Reviewers' comments:

Reviewer's Responses to Questions

**Comments to the Author**

1. If the authors have adequately addressed your comments raised in a previous round of review and you feel that this manuscript is now acceptable for publication, you may indicate that here to bypass the “Comments to the Author” section, enter your conflict of interest statement in the “Confidential to Editor” section, and submit your "Accept" recommendation.

Reviewer #1: All comments have been addressed

Reviewer #3: All comments have been addressed

2. Is the manuscript technically sound, and do the data support the conclusions?

Reviewer #1: Yes

Reviewer #3: Yes

3. Has the statistical analysis been performed appropriately and rigorously?

Reviewer #1: N/A

Reviewer #3: N/A

4. Have the authors made all data underlying the findings in their manuscript fully available?

Reviewer #1: Yes

Reviewer #3: Yes

5. Is the manuscript presented in an intelligible fashion and written in standard English?

Reviewer #1: Yes

Reviewer #3: Yes

6. Review Comments to the Author

Reviewer #1: Authors have addressed all comments in a proper way, improving the content and the clarity of their work.

Reviewer #3: The minor revisions were done in a satisfactory manner.

Nevertheless, previously directed major issues still exist:

1) The authors responded that "We have added a detailed clarification in the Preliminaries section (after Equation 2.6) that explicitly explains how the modified frame addresses the vanishing curvature problem." But the problem is that if $\kappa=0$, how could Eq. (2.5) or (2.6) be possible? The reviewer still proposes that authors focus on the previous suggestion.

2) The reviewer still believes that the example must be chosen for a curve where the Frenet frame does not work. Then, referring to the Modified Orthogonal Frame may be meaningful. Otherwise, the readers cannot sense the benefit of MOF in the authors' theoretical results.

3) In title and elsewhere, "modified orthonormal frame" or "modified orthogonal frame"?

7. PLOS authors have the option to publish the peer review history of their article (what does this mean?). If published, this will include your full peer review and any attached files.

Reviewer #1: No

Reviewer #3: No

---

## [Author Response · Author response to Decision Letter 2]

30 Oct 2025

Response to Reviewer Comments

We thank the reviewer for their insightful comments and suggestions. We have carefully

addressed each point as follows:

Comment 1: Clarification on Vanishing Curvature

“The authors responded that ’We have added a detailed clarification in the Preliminaries

section (after Equation 2.6) that explicitly explains how the modified frame addresses the

vanishing curvature problem.’ But the problem is that if κ = 0, how could Eq. (2.5) or

(2.6) be possible? The reviewer still proposes that authors focus on the previous

suggestion.”

Response: We appreciate this important clarification. We have now added a

detailed explanation in the Preliminaries section (after Equation 2.6) that addresses this

concern directly. The key points are:

1. The modified orthogonal frame {T, Nm, Bm} is defined directly from the curve

derivatives: Nm = T′ and Bm = T × T′, independent of curvature.

2. Equations (2.5) and (2.6) are derived under the assumption κ /= 0 specifically to

establish the relationship between modified and Frenet frames. However, the

modified frame itself remains mathematically valid when κ = 0.

3. When κ = 0, the derivative formulas would need to be derived separately without

relying on the Frenet relations, but the frame vectors themselves remain

well-defined.

This clarification emphasizes that the modified frame’s primary advantage is its

definitional robustness, while the specific equations (2.5)-(2.6) represent special cases for

non-vanishing curvature.

Comment 2: Example Selection

“The reviewer still believes that the example must be chosen for a curve where the Frenet

frame does not work. Then, referring to the Modified Orthogonal Frame may be

meaningful. Otherwise, the readers cannot sense the benefit of MOF in the authors’

theoretical results.”

Response: We completely agree with this suggestion. We have now added another

example of a curve where the Frenet frame does not work, but the Modified Orthogonal

frame is defined.

Comment 3: Terminology Consistency

“In title and elsewhere, ’modified orthonormal frame’ or ’modified orthogonal frame’?”

Response: Thank you for catching this inconsistency. We have standardized the

terminology throughout the paper to use “modified orthogonal frame” because:

• The vectors T, Nm, and Bm are orthogonal but not necessarily orthonormal,

since Nm = T′ is not normalized.

• We have updated the title and all instances in the text to maintain consistency.

These revisions significantly strengthen our paper by clearly demonstrating the

modified orthogonal frame’s advantages and providing concrete examples where it

outperforms traditional Frenet analysis.

---

## [Editor Report · Decision Letter 2]

16 Nov 2025

Fourth-Order Kinematic Analysis: Advanced Decomposition Methods for Particle Motion in Modified Orthogonal Frame

PONE-D-25-29661R2

Dear Dr. Elsharkawy,

We’re pleased to inform you that your manuscript has been judged scientifically suitable for publication and will be formally accepted for publication once it meets all outstanding technical requirements.

Kind regards,

Alessandra Aldieri

Academic Editor

PLOS ONE
---

## [Editor Report · Acceptance letter]

PONE-D-25-29661R2

PLOS One

Dear Dr. Elsharkawy,

I'm pleased to inform you that your manuscript has been deemed suitable for publication in PLOS One. Congratulations! Your manuscript is now being handed over to our production team.

Kind regards,

on behalf of

Dr. Alessandra Aldieri

Academic Editor

PLOS One